# CCND1 Overexpression in Idiopathic Dilated Cardiomyopathy: A Promising Biomarker?

**DOI:** 10.3390/genes14061243

**Published:** 2023-06-10

**Authors:** Khatereh Dehghani, Agata Stanek, Arash Bagherabadi, Fatemeh Atashi, Mohammad Beygi, Amirreza Hooshmand, Pezhman Hamedi, Mohsen Farhang, Soghra Bagheri, Samaneh Zolghadri

**Affiliations:** 1Department of Cardiology, Jahrom University of Medical Sciences, Jahrom 7414846199, Iran; 2Department and Clinic of Internal Medicine, Angiology and Physical Medicine, Faculty of Medical Sciences in Zabrze, Medical University of Silesia, Batorego 15 Street, 41-902 Bytom, Poland; 3Department of Biology, Faculty of Sciences, University of Mohaghegh Ardabili, Ardabil 5619911367, Iran; 4Faculty of Medicine, Jahrom University of Medical Sciences, Jahrom 7414846199, Iran; 5Department of Agricultural Biotechnology, College of Agriculture, Isfahan University of Technology, Isfahan 8415683111, Iran; 6Department of Molecular and Cellular Sciences, Faculty of Advanced Sciences & Technology, Tehran Medical Sciences, Islamic Azad University, Tehran 1916893813, Iran; 7Research Center, Department of Medical Laboratory Sciences, Faculty of Medicine, Jahrom University of Medical Sciences, Jahrom 7414846199, Iran; 8Molecular Study and Diagnostic Center, Jahrom University of Medical Sciences, Jahrom 7414846199, Iran; 9Medical Biology Research Center, Health Technology Institute, Kermanshah University of Medical Sciences, Kermanshah 6714415185, Iran; 10Department of Biology, Jahrom Branch, Islamic Azad University, Jahrom 7414785318, Iran

**Keywords:** gene network analysis, microRNA, bioinformatics, diabetes, DisGeNET

## Abstract

Cardiomyopathy, a disorder of electrical or heart muscle function, represents a type of cardiac muscle failure and culminates in severe heart conditions. The prevalence of dilated cardiomyopathy (DCM) is higher than that of other types (hypertrophic cardiomyopathy and restrictive cardiomyopathy) and causes many deaths. Idiopathic dilated cardiomyopathy (IDCM) is a type of DCM with an unknown underlying cause. This study aims to analyze the gene network of IDCM patients to identify disease biomarkers. Data were first extracted from the Gene Expression Omnibus (GEO) dataset and normalized based on the RMA algorithm (Bioconductor package), and differentially expressed genes were identified. The gene network was mapped on the STRING website, and the data were transferred to Cytoscape software to determine the top 100 genes. In the following, several genes, including *VEGFA*, *IGF1*, *APP*, *STAT1*, *CCND1*, *MYH10*, and *MYH11*, were selected for clinical studies. Peripheral blood samples were taken from 14 identified IDCM patients and 14 controls. The RT-PCR results revealed no significant differences in the expression of the genes *APP*, *MYH10*, and *MYH11* between the two groups. By contrast, the *STAT1*, *IGF1*, *CCND1*, and *VEGFA* genes were overexpressed in patients more than in controls. The highest expression was found for *VEGFA*, followed by *CCND1* (*p* < 0.001). Overexpression of these genes may contribute to disease progression in patients with IDCM. However, more patients and genes need to be analyzed in order to achieve more robust results.

## 1. Introduction

Cardiovascular diseases (CVDs) are among the leading causes of mortality and morbidity globally, representing a considerable burden to families and society [1]. As statistics indicate, the prevalence of CVDs has virtually doubled during the past two decades, and, worryingly, deaths from CVDs have risen from 12.1 to 18.6 million throughout this period [1]. Among all CVDs, cardiomyopathies are the most common and represent cardiac muscle failure, culminating in serious heart conditions. Dilated cardiomyopathy (DCM) refers to heart muscle failure characterized by ventricular enlargement, systolic dysfunction [2], and impaired contraction, resulting in a lessened ability to pump blood [3]. Most contributors to DCM are genetic and acquired factors, such as mutations, infections, autoimmunity, metabolic or endocrine dysfunction, exposure to toxins, neuromuscular diseases, and pregnancy [4].

Idiopathic dilated cardiomyopathy (IDCM) is a type of DCM for which the underlying causes are unclear [5]. However, genetics play a crucial role in this disease, and many genes have contributed to its progression. In a study among African Americans, the IDCM heritability was 33%. In this ethnic group, a variant in a novel intronic locus in the gene *CACNB4* contributes to IDCM [6]. In another study, *IL1RL1*, *TIMP-1*, and *TIMP-4* have been reported as similarly altered markers in adult and pediatric IDCM hearts. Furthermore, microRNAs 29a–c are substantially downregulated in pediatric IDCM patients [7].

Given the high prevalence of IDCM and the mortality caused by it, it is becoming increasingly critical to identify high-risk patients at earlier stages so as to start therapy before the late phase of the disease [8]. Numerous studies have examined the data to identify the best screening and prognostication indices [9], but they have been inconclusive. Currently, a direct biopsy of the myocardium achieves the most exact diagnosis, but it is invasive. As an alternative, the peripheral blood sample provides a non-invasive diagnosis [10]. In fact, more than 84% of genes expressed in heart tissue overlap with genes expressed in blood cells [11], and peripheral blood gene expression profiles have the potential to identify genetic signatures associated with cardiovascular diseases [10,12,13,14,15]. Particularly, it has been shown that gene expression changes in the peripheral blood of IDCM patients can reflect changes in left ventricular function [16]. In addition, several reports have indicated that gene expression alterations in blood cells occur in the early stages of the disease and before the condition worsens [17,18,19,20]. Considering that aberrant molecular variations and environmental modifications leading to cardiovascular disease occur before tissue morphological abnormalities [21], the study of molecular variations can serve as a valuable tool for the early diagnosis of asymptomatic patients. Moreover, numerous studies have reported that gene expression changes in peripheral blood cells are associated with coronary artery disease severity [22,23,24], and gene expression analysis is a practical method capable of reflecting genetic predisposition, disease activity, environmental modifier effects, and therapeutic responses [23,24,25,26,27]. In addition to its scientific value, the blood gene expression profile offers a cost-effective and less-invasive alternative to invasive measurements [10]. Therefore, in this study, by analyzing the gene expression in cardiac tissues of patients who were in advanced stages of the disease (GSE5406 dataset), we determined the genes that have high expression in the advanced stages of the disease. Afterwards, we used blood samples (the present study) to know which of these advanced-stage genes have increased expression in the early stages of the disease (suitable for introduction as biomarkers and prognosticators).

In this research, we conducted a comprehensive bioinformatics analysis, including the identification of DEGs, gene ontology (GO) enrichment, and the construction of a PPI network. Furthermore, we validated the expression of several genes identified through bioinformatics analysis as having increased expression in IDCM patients using the RT-PCR method. It is important to note that our study differs from previous research in that we employed distinct thresholds and parameters for identifying DEGs and conducted multiple rounds of analysis to ensure the robustness of our findings.

## 2. Materials and Methods

### 2.1. Data Collection

The gene expression profile dataset GSE5406 was downloaded from the National Center for Biotechnology Information (NCBI) and the Gene Expression Omnibus (GEO) database (http://www.ncbi.nlm.gob/geo/ (accessed on 10 August 2022)). The dataset contained 210 RNA samples, as follows: 108 cases of human explanted left ventricular (LV) myocardial with systolic heart failure (HF) due to ischemic cardiomyopathy (IC) (IC-HF), 86 cases of explanted LV myocardial with systolic HF due to idiopathic dilated cardiomyopathy (IDC) (IDC-HF), and 16 non-failing controls with a normally functioning LV myocardium from the unused donor heart. Based on the report by Hannenhalli et al. [28], all heart failure patients had New York Heart Association class 3 to 4 symptoms and LV systolic dysfunction, with a mean ± SD ejection fraction of 14 ± 8%. Nonfailing controls had normal left ventricular function with a mean ejection fraction of 56 ± 7% (*p* = 0.0001 versus failing). Ages were comparable in subjects with heart failure (57 ± 12 years) and nonfailing controls (54 ± 12; *p* = 0.3) [28]. Further information on sample characteristics was not reported. For further studies, 102 samples of IDC-HF and non-failing controls were selected in total. The data were generated using an Affymetrix Human Genome U133A Array (HG-U133A). In addition, GPL96 platform annotation was performed to map gene probes to gene names.

### 2.2. Data Preprocessing

Series matrix files and associated annotations for the dataset were acquired from the GEO database [29] using the R programming language package “GEOquery” (https://www.r-project.org/ (accessed on 1 August 2022)). All arrays were pre-processed and normalized together using the robust multichip average (RMA) method (http://www.bioconductor.org/). For those multiple expression values that corresponded to an individual gene symbol, a maximum level was adopted.

### 2.3. DEG Identification

Empirical Bayes statistics (eBayes) were used via the Linear Models for Microarray Data “limma” package [30] of R to uncover the DEGs in the IDC-HF samples compared to the normal LV samples. DEGs were selected with the following cut-off:|log_2_ FC| > 0.3 and *p*-value < 0.05. To avoid gene name corruption in Microsoft Excel, the Escape Excel [31] plug-in was implemented. Using the “pheatmap” package, a hierarchical clustering heatmap for DEGs was plotted.

### 2.4. Enrichment and Functional Analysis

The R package “ClusterProfiler” [32] was used to carry out KEGG pathway analysis [33] and GO enrichment analysis [34]. The cut-off was set at an adjusted *p*-value < 0.05 (obtained using the BH procedure), and this work was conducted to discover the related fundamental processes and biological pathways.

### 2.5. PPI Network Construction and HUB Detection

Utilizing the Search Tool for the Retrieval of Interacting Genes (STRING, https://stringdb.org/, accessed on 15 August 2022) v11.5 database [35], the PPI network was built to investigate the interactions between upregulated genes in IDC-HF. Only interactions with a medium confidence of a combined score > 0.4 were acquired. Moreover, Cytoscape (v3.8.2) [36] was used to evaluate network parameters for further HUB identification. All topological parameters were computed using the “CentiScaPe 2.2” plug-in.

### 2.6. Sampling

Fourteen IDCM patients, aged 45–55 years, comprising seven men and seven women, referred to healthcare centers affiliated with Jahrom University of Medical Sciences (Jahrom, Iran), were investigated against fourteen controls with the same age and gender distribution. Patients were diagnosed with IDCM by echocardiography using visual and Simpson methods. All patients underwent coronary angiography (CAG) to reject HF ischemic and other secondary causes. Patients were clinically stable for at least a month after receiving diuretics, angiotensin-converting inhibitors, digoxin, and β-blockers. Inclusion criteria included having a left ventricular ejection fraction < 45% (LVEF < 45%) and undergoing left ventricular catheterization (LVC) and CAG with normal results or minimal coronary artery involvement. Exclusion criteria were having CVDs, significant coronary artery stenosis (> or =50%), a prior history of MI, severe heart valve disease, restrictive or hypertrophic cardiomyopathy, prolonged or uncontrolled systemic diseases such as acute and chronic cardiac involvement, myocarditis, thyroid disease, drug abuse, HIV, chronic kidney dysfunction, and consuming drugs with myocardial toxicity. The patients did not report any familial history of DCM. All the participants were informed of the study procedures and asked to sign a consent form before entering the study. Whole blood samples (10 mL from each participant) were obtained and stored at −80 °C for further examinations. The JUMS’s research council approved the study under the ethical code IR.JUMS.REC.1398.122.

### 2.7. Total RNA Isolation, cDNA Synthesis, and RT-PCR Validation

Whole blood samples were collected into ethylenediaminetetraacetic acid (EDTA) collection tubes. Peripheral blood mononuclear cells were isolated by histopaque-ficoll (SIGMA) centrifugation. Total RNA was isolated using Invitrogen™ TRIzol™ Reagent (Kimia Gostar Pooyesh Co., Ltd., Tehran, Iran). The RNA purity and quantity were estimated by a colorimetric assay. The OD values at 260, 280, and 320 nm were read using a UV/Vis spectrophotometer (Eppendorf, Germany), and the RNA concentration was measured as *RNA.Conc.* = (*OD*_260_ − *OD*_280_) × 40 × 100. The 260/230 absorbance ratio was 1.8, indicating the acceptable purity of the RNA extracted. cDNA was constructed using random hexamer primers and the Fermentas cDNA Synthesis Kit (Sankt Leon-Rot, Germany), which has RevertAid H Minus Reverse Transcriptase. This enzyme has a point mutation, enabling it to inhibit RNase H activity entirely. It further prevents RNA decomposition and allows the synthesis of a full-length cDNA from the primary strand. Additionally, the kit contains Ribolock RNase Inhibitor, which prevents RNA decomposition up to 55 °C. Primers were designed according to PREMIER Biosoft International Inc. (San Francisco, CA, USA) (Table 1). All the primers were investigated for the possibility of the formation of secondary structures, primer-dimer (PD) formation during the PCR, primer melting temperature (Tm), and proper Tm ranges. The mRNA sequences of target and housekeeping genes were obtained from NCBI, and all primers were precisely investigated for exact attachment to the sequences obtained. The NCBI Blast tool was used to investigate the likely primer attachment to other sequences.

The RT-PCR was carried out in an ABI thermocycler (ABI Co., Chiyoda City, Tokyo) using the 2^–∆∆Ct^ method. The reaction was performed using SYBR^®^ Green Fluorescent as a marker. The Ct:Ct values for all eight target genes and the housekeeping gene (GAPDH) were estimated from RT-PCR graphs. Similarly, ∆∆Ct was obtained by subtracting the ∆Ct of each state from the ∆Ct of controls. Furthermore, the relative increase in gene expression level was calculated using the 2^–∆∆Ct^ method.

### 2.8. Over-Representation ANALYSIS

We utilized the “DOSE” package in R [37] to apply over-representation analysis for the disease ontology [38] and DisGeNET [39] databases using enrichDO and enrichDGN commands, respectively. We set the cut-off at an adjusted *p*-value < 0.05 (obtained using the BH procedure). This approach led us to find significant gene-disease associations among our validated hub genes.

## 3. Results

### 3.1. Data Preprocessing and Identification of DEGs

The GSE5406, a microarray expression dataset, includes mRNA expression data from 102 samples, 86 IDCM samples, and 16 normal LV samples. For our analysis, we selected IDCM and normal samples and downloaded the series matrix file, which was pre-processed and normalized together using the robust multichip average (RMA) method. Using the limma R program, the DEGs were filtered (criteria: |log_2_ FC| > 0.3 and *p*-value < 0.05). Finally, 1287 DEGs (Appendix A) were extracted from the IDCM samples and compared to the normal left ventricle function samples, comprising 632 upregulated and 655 downregulated DEGs (Figure 1 depicts the heatmap for upregulated and downregulated DEGs).

### 3.2. Differentially Expressed Gene Enrichment Analysis

We performed KEGG pathway analysis by using the ClusterProfiler package for upregulated DEGs. KEGG analysis demonstrated that human papillomavirus infection, viral myocarditis, focal adhesion, ECM receptor interaction, and the calcium signaling pathway were linked to our DEGs (Table 2). The top 20 enriched KEGG pathways are shown in Figure 2A.

Furthermore, GO analysis was performed by using the ClusterProfiler package for upregulated DEGs. The result revealed that the most significant GO terms under the biological processes category (in descending order of adjusted *p*-value) were extracellular matrix organization (GO:0030198), extracellular structure organization (GO:0043062), and external encapsulating structure organization (GO:0045229). Furthermore, the most significant GO terms under the molecular functions category were extracellular matrix structural constituent (GO:0005201), glycosaminoglycan binding (GO:0005539), and integrin binding (GO:0005178). Finally, the most significant GO terms under the cellular component category were collagen-containing extracellular matrix (GO:0062023), contractile fiber (GO:0043292), and Z disc (GO:0030018) (Figure 2B).

### 3.3. PPI Network Construction and HUB Selection

For this article, we selected the most upregulated DEGs and used the STRING online database to produce a protein-protein interaction (PPI) network. According to the STRING analysis, our network consisted of 615 nodes and 2543 edges. The top 100 genes, ranked based on the intersections between four topological algorithms (i.e., degree, betweenness, closeness, and centroid), were expected to play a significant role in biological processes and were set as HUBs (Figure 3, Appendix A). Although in vivo examination and validation of all these genes can be of particular importance in understanding the pathogenesis of IDCM, due to experimental limitations, we selected seven of the top 100 genes (nominated as “HUB of HUBs”) based on the bioinformatics analysis results and the literature review. In fact, one of the common methods for selecting genes (i.e., seven HUB of HUBs in this study) out of the top ones in the PPI network (to introduce them as biomarkers or potential therapeutic targets and consider further experimental studies) is the results of patient survival analysis. Despite the abundance of survival data for cancer patients, the absence of such information for DCM patients presents a technical limitation. Consequently, we relied on an extensive literature review as an alternative and considerable method for gene selection. Thus, due to these limitations, five genes (*VEGF-A*, *STAT1*, *APP*, *CCND1*, and *IGF1*) with the best betweenness ranking were selected based on the bioinformatics analysis, and two genes related to heart disease were selected according to the literature review (*MYH 10* and *MYH 11*): 1. Vascular endothelial growth factor-A (*VEGF-A*) can regulate angiogenesis, vascular permeability, and inflammation. Based on our bioinformatics analysis, this gene was among the top ten genes, but there were few documents and pieces of evidence examining its expression in IDCM patients. 2. Signal transducer and activator of transcription 1 (*STAT1*) is a member of the *STAT* family, and it has been proposed that *STAT1* promotes the generation of larger infarcts, which can lead to heart failure, by enhancing apoptosis and negatively regulating autophagy [40]. 3. The amyloid precursor protein (*APP*) is known as a precursor protein of Alzheimer’s disease (AD)-related amyloid β-protein (Aβ) [41]. Recent studies identified Aβ aggregates in the hearts of patients with dilated cardiomyopathy [42]. It was important and exciting for us to examine the expression of *APP* in patients without a history of Alzheimer’s. 4. *CCND1* is an important cell cycle gene involved in three pathways (the cell cycle, Hedgehog (Hh) signaling, and Wnt signaling). Limited previous observations have reported its upregulation in IDCM [43,44]. 5. Insulin-like growth factor-1 (*IGF-1*) is a peptide hormone that activates canonical and non-canonical signaling pathways in the heart. It has a direct growth-promoting effect on cardiomyocyte hypertrophy. 6. Myosin heavy chain (*MYH10*), the only non-muscle myosin, is expressed in the heart and required for normal heart development. Recently, the upregulation of *MYH10* in the cardiomyocytes of adults with heart failure has been reported [45]. 7. Myosin heavy chain 11 (*MYH11*) belongs to the *MYH* family and is a myocardial contractile protein. Hydrolyzed proteins are involved in muscle contraction through adenosine triphosphate. Myocardial cytoskeleton proteins are important for maintaining the structural and functional integrity of the myocardium [46]. However, there are few studies on the changes in cytoskeletal proteins associated with dilated cardiomyopathy. Figure 4 represents the violin plots of selected HUB of HUBs expression patterns.

### 3.4. RNA Isolation and Quantification, cDNA Synthesis, and RT-PCR

Isolated RNA was quantified with NanoDrop (Biotech. ENG). The OD 260/280 and OD 260/230 ratios of 1.8 to 2 and 1.7 to 2 indicated acceptable RNA quantities. The synthesized cDNA was similarly quantified with NanoDrop (Biotech. ENG). The melting curve was investigated at 60 to 96 °C. The highest expression in patients compared to controls was found for the gene *VEGFA* (*p* < 0.001), followed by the genes *CCND1* (*p* < 0.001) and *IGF1* (*p* = 0.001). *STAT1* was also overexpressed in patients (*p* = 0.017). Contrarily, the genes *APP* (*p* = 0.823), *MYH11* (*p* = 0.781), and *MYH10* (*p* = 0.502) were not significantly more expressed in patients compared to controls (Figure 5).

### 3.5. Over-Representation Analysis for the DisGeNET and DO

Over-representation analysis (ORA) on four significantly expressed HUBs of HUBs (*VEGFA*, *IGF1*, *CCND1*, and *STAT1*) was applied by the DOSE package for two databases of the disease gene network (DisGeNET) and disease ontology (DO). However, it is essential to interpret these results cautiously and consider the databases’ constraints. DisGeNET integrates data from expert-curated repositories, scientific literature, animal models, and GWAS catalogs. Nevertheless, it is important to recognize that the data contained in these databases may have inherent limitations, such as errors in text mining or insufficient annotations.

In this study, three keywords were used to filter out diseases that are related to this project: “cardiomyopathy,” “heart failure,” and “ischemic.” Some genes were also involved in “cardiovascular” diseases, but this keyword was ignored to prevent generalization. According to the DisGeNET and DO analyses, two genes *(IGF1* and *VEGFA)* were significantly related to IDCM, while there is no link between *STAT1* and *CCND1* with IDCM (Figure 6, Table 3).

It should be noted that *CCND1*’s role was detected by DisGeNET in two related diseases. The former was “Cardiomyopathy, Familial Idiopathic,” while after studying the stated PMIDs, it was found that all of them were related to “invasive ductal carcinoma (IDC)” incorrectly, instead of idiopathic dilated cardiomyopathy (IDC), due to possible errors in text mining. The latter was “myocardial ischemia,” given only one PMID, which was not enough to conclude the role of *CCND1* in that disease. To sum up, there is insufficient cardio-relational experimental evidence for *CCND1*. However, it is important to emphasize that the absence of significant associations in the current analysis does not preclude the existence of a potential relationship between CCND1 and IDCM. A deeper exploration of the potential involvement of CCND1 in IDCM is needed through rigorous experiments and validation.

## 4. Discussion

DCM is a consequence of various pathogenic factors, such as genetic, infectious, hormonal, and environmental factors [47]. However, knowledge about the genotype-phenotype relationship is still unknown. Therefore, DCM is almost always diagnosed late, which in turn causes a poor prognosis. In recent years, gene sequencing and bioinformatics methods have resulted in new ideas for understanding the mechanisms of disease development, disease diagnosis, and personalized precision medicine development and presented some meaningful results. Sun and Li (2023) used WGCNA and the machine learning algorithm to identify biomarkers of IDCM and obtained two hub genes, *AQP3* and *CYP2J2*, which have the potential to serve as targets for the diagnosis and management of IDCM [48]. Liu et al. (2022) investigated the candidate genes and pathways involved in DCM patients (data sets GSE3585 and GSE5406) and predicted the microRNAs (miRNAs) targeting the hub genes. Their screening criteria were set as *p* < 0.05 and |log fold change (FC)| > 0.589. Furthermore, they investigated the pattern of immune cell infiltration in DCM. The top 10 hub genes included collagen type III alpha 1 chain (COL3A1), COL1A2, signal transducer and activator of transcription 3 (STAT3), C-C motif chemokine ligand 2 (CCL2), fibromodulin (FMOD), aspirin (ASPN), C-X-C motif chemokine ligand 12 (CXCL12), lumican (LUM), heat shock protein 90 alpha family class A member 1 (HSP90AA1), and osteoglycin (OGN) [49]. Si (2023) has screened the associated genes and biological pathways of inflammatory dilated cardiomyopathy (DCMi) [50]. Luo et al. (2020) identified novel long non-coding RNA (lncRNA) biomarkers associated with DCM and revealed the potential molecular mechanisms of DCM development using bioinformatics approaches (dataset GSE5406 from GEO) [51]. Zhang et al. (2022) screened diagnostic biomarkers and identified the landscape of immune infiltration in DCM (dataset GSE141910 from Geo). They found that ASPN, CD163, IL10, and LUM could predict the occurrence of DCM [52]. Huang et al. (2018) found that Fos proto-oncogene, AP-1 transcription factor subunit, tissue inhibitor of metalloprotease-1, and serpin family E member 1 may serve as therapeutic targets in DCM [25].

In this study, we investigated which of the DEGs related to the advanced stages of the disease (confirmed by bioinformatics methods using the GSE5406 dataset) have increased expression in the early stages of the disease (using the RT-PCR method in blood samples of IDCM patients participating in this study). It is noteworthy that the dataset used in this study (GSE5406) has already been analyzed by several researchers to predict potential targets [25,53], but in this study we set different thresholds for some parameters compared to previous studies. For instance, a similar paper for this dataset used |log2 FC| > 0.589 to detect DEGs, while we used |log2 FC| > 0.3. FC is a main parameter to show the change in expression levels of genes. FC analysis is a very intuitive method to identify DEGs [54]. Although in most cases, |log2 FC| is set to ≥1, sometimes these cutoff values can significantly change the interpretation of the microarray due to the deletion of some hub genes [55]. For instance, a minor change in transcription factor expression may alter the entire system and induce a high effect across the network of genes [56]. Thus, we covered more genes in our study that could possibly have significant impacts on cellular processes while having a minor gene expression level. Furthermore, we took additional steps to ensure the reliability of our findings and performed multiple rounds of analysis with different thresholds and parameters (e.g., different topological centrality parameters for detecting HUBs in our PPI network and then selecting HUBs of HUBs) and used different software packages for the analysis to confirm the robustness of our results (e.g., using DOSE and enrichplot packages to perform and visualize over-representation analysis (ORA) on significantly expressed HUBs of HUBs). Finally, from the DEGs identified by bioinformatics methods, seven genes were selected to check their expression in the blood of patients participating in this study.

RT-PCR tests showed that four of the seven selected genes (*VEGFA*, *IGF1*, *CCND1*, and *STAT1*) have increased expression in blood samples from patients. Some studies, in accordance with the results of our RT-PCR experiments, showed an increase in the expression of some genes in a variety of heart diseases. However, for genes whose expression changes were not observed in our study, we did not find any reports on the expression of these genes in patients with IDCM or other heart diseases. The literature review revealed that, in general, there is little information on the expression changes of the HUB genes selected in this study in IDCM patients.

*VEGFA* is one of the HUB genes whose increased expression in patients was observed in our study. The generation of six isoforms, including *VEGF111*, *VEGF121*, *VEGF145*, *VEGF165*, *VEGF189*, and *VEGF206*, from *VEGFA* mRNA under the alternative splicing process has been reported [57]. *VEGFA* and its two receptors, *VEGFR-1* and *VEGFR-2*, play fundamental roles in angiogenesis under physiological and pathological conditions, as well as in neurological and cardiac growth and morphogenesis [46,58]. Disturbance in angiogenesis has been shown to contribute to the development of myocardial interstitial fibrosis in IDCM [59,60,61,62]. Some evidence has confirmed that serum *VEGF-A* [61] or total *VEGF* [60] is significantly higher in patients with IDCM compared to controls. Moreover, significant increases in serum *VEGF* levels were reported in DCM patients compared to ischemic cardiomyopathy patients [63]. However, another study showed that, in the heart tissue of patients with end-stage DCM, the mRNA transcript levels of *VEGF165* and *VEGF189* and the protein level of *VEGF-A* decreased compared to controls [64]. In fact, it has been indicated that in the disease’s acute stage, the expression of genes and receptors may be altered significantly, with varying intensities and at different times. Accordingly, the concentration of *VEGF-A* may fluctuate significantly [65]. In this regard, it has been reported that *VEGF-A* is strongly expressed in the acute and subacute stages of CNS damage, but its expression decreases over time [66].

*CCND1* was one of the HUB genes that our RT-PCR studies confirmed to be overexpressed in IDCM patients. *CCND1* is a proto-oncogene that manages progression via the G1-S phase of the cell cycle [67], and its involvement in heart diseases has been reported [68]. In a study reported on children and adults with IDCM in the end-stages, it was shown that the *CCND1* gene is not overexpressed in adults, but it is overexpressed in children [42]. A significant association of microRNAs (miRNAs) with heart diseases such as cardiac arrhythmia, myocardial infarction, and cardiac hypertrophy has been confirmed [69]. For instance, it has been revealed that several miRNAs (e.g., miR-34a, miR-28, miR-148, and miR-93) are differentially expressed in patients with left ventricular hypertrophy (LVH) and normal controls [70]. Moreover, more recent studies confirmed that miR-93 is reduced in LVH patients and that *CCND1* is the direct target of miR-93 [68]. In mouse models, the *MYCN* gene has been revealed to induce cardiomyocyte proliferation, at least in part by upregulating *CCND1*, *CCND2*, and the inhibitor of DNA binding 2 (ID2). *MYCN* is a multifunctional transcription factor that plays an essential role in the development of disease [71]. *MYCN* has been indicated to be the major disease-causing gene for Feingold syndrome, a developmental disorder characterized in part by congenital heart abnormalities [72]. However, some other reports showed that the upregulation of *CCND1* and *CCND2* suppressed DCM caused by *TTN* gene insufficiency [73]. Truncating variants of *TTN* are common in patients with IDCM [74,75,76]. Titin, encoded by the gene *TTN*, is a giant sarcomeric protein that is critical for cardiac contraction and relaxation [77].

*IGF-1* is a peptide hormone homologous to pro-insulin, which is expressed in most tissues and is involved in cell proliferation, apoptosis, migration, and differentiation [78]. The results of epidemiological studies on the relationship between *IGF-1* levels and the risk of cardiovascular disease were inconsistent [79], as various reports indicated that both an increase and decrease in serum *IGF-1* levels were involved in heart disease [80]. Similarly, a meta-analysis conducted in 2015 showed that both low and high *IGF-1* levels were associated with an increased risk of cardiovascular disease [79].

Indeed, more recent studies have also not reached a consensus on the association of *IGF-1* with heart disease (including dilated cardiomyopathy): while some have reported a decrease in *IGF-1*, others have reported an increase, and some studies have not even found an association between *IGF-1* levels and heart failure. In this regard, some evidence associates the decline in *IGF-1* expression with heart disease. For instance, in a cohort study of 337 patients, *IGF-1* was reported as a predictor of cardiovascular mortality in patients with heart failure, so older patients with lower serum *IGF-1* levels showed higher mortality [81]. Another report described that *IGF-1* signaling was inhibited in end-stage DCM [82]. Moreover, a study on DCM rats revealed that the intramuscular injection of human umbilical cord-derived mesenchymal stem cells (hUCMSC) improved cardiac function, and the same study suggested that the expression of *IGF-1*, *HGF*, and *VEGF* in the myocardium of DCM rats was remarkably increased by hUCMSC injection [83]. In contrast, other evidence suggests that increased *IGF-1* expression is involved in heart disease. Free IGF-1 levels have been shown to be higher in patients with myocardial infarction compared to controls [64]. Furthermore, *IGF-1* levels in patients with HFpEF (heart failure with preserved ejection fraction) were significantly higher than the *IGF-1* levels of their HFrEF (heart failure with reduced ejection fraction) counterparts [84]. Moreover, it was shown that the concentration of *IGF*-binding protein *(IGFBP)-1* and the ratio of *IGFBP-1/IGF-1* in patients with heart failure were significantly lower than in a control group, and they can be used to easily identify patients with and without heart failure [85]. Due to the fact that *IGFBP1* and *IGFBP2* have inhibitory effects on the biological activity of *IGF-1*, it can be concluded that, in this group of patients, although the amount of *IGF-1* did not increase due to the decrease in the inhibitory effect of its binding proteins, in fact, *IGF-1* had increased activity [86]. Furthermore, some evidence suggests that elevated *IGF-1* levels may be causally correlated with a higher risk of type 2 diabetes [68], and another report indicated that patients with type 2 diabetes were notably associated with an increased risk of IDCM in all age and sex categories, with the exception of men over 64 years of age [87]. On the other hand, some studies have linked an increase in *IGF-1* system activity to a decrease in lifespan [88], while other reports have revealed that an increase in *IGF-1* is associated with increased survival in people over 65, unlike younger people [89]. In general, it can be concluded that an increase in *IGF-1* in the under-65 population can be dangerous, which is highly consistent with the data of our study, in which the subjects were under 65 years of age.

*STAT1* is another gene whose high expression was observed in the IDCM patients who participated in our study. *STAT1*, encoding a signal transducer and activator of transcription factor, is involved in apoptosis and the interferon response [90]. Some reports have suggested that this gene is involved in dilated cardiomyopathy. It has been shown that mutations of sodium voltage-gated channel alpha subunit 5 (*SCN5A*) are involved in dilated cardiomyopathy [91,92]. Furthermore, mice with the *SCN5A N1325S* mutation in the heart (TG-NS mice) exhibited the phenotype of dilated cardiomyopathy and heart failure [93], and other studies have indicated the upregulation of *STAT1* in the hearts of TG-NS mice [90]. Moreover, in vitro studies on engineered myocardial tissues stated that interferon-γ (*IFN-γ*) damages myofibrillar organization and contractile force production in human cardiomyocytes through upregulation of the *JAK/STAT* signaling pathway and downregulation of multiple sarcomeric proteins. In fact, circulating *IFN-γ* (a pro-inflammatory cytokine) is enhanced in numerous clinical conditions, including acute coronary syndrome, autoimmune and inflammatory diseases, sepsis, and viral infections, which are correlated with a high risk of myocardial dysfunction [94]. In addition, *Trypanosoma cruzi*, a protozoan parasite, causes zoonotic Chagas disease, which is a chronic and systemic infection that typically brings about progressive dilated cardiomyopathy and gastrointestinal manifestations [95]. Some in vitro studies revealed that in response to *T. cruzi*, intracellular *STAT1* was increased both at the mRNA and protein levels [96].

Further analysis was performed through the DisGeNET database to find the correlation between four significant HUB of HUBs genes (*VEGFA*, *IGF1*, *CCND1*, and *STAT1*) and IDCM. According to DisGeNET data, confirmatory findings were reported for two genes (*IGF1* and *VEGFA*) with an IDCM link in DisGeNET. These results suggest that *IGF1* and *VEGFA* are also associated with other heart diseases similar to IDCM. Furthermore, DisGeNET data show that two genes, including *STAT1* and *CCND1*, have not been associated with IDCM, but *STAT1* is related to congestive heart failure. Although, based on DisGeNET results, *CCND1* is not related to IDCM, its upregulation in IDCM has been previously reported in children [44]. The same report has shown that *CCND1* is not overexpressed in adults at the end stage of IDCM [44]. Thus, our study is the first evidence of *CCND1*’s upregulation in IDCM adult patients at an early stage, and *CCND1* might be a potential newfound gene for cardiomyopathy.

In contrast to the genes that exhibited increased expression in the patients who participated in our study, some upregulated genes (based on analysis of GEO data) did not show significant changes in the patients compared to the control group, based on the RT-PCR results. These genes included *MYH10*, *MYH11*, and *APP*. There are several points to be made regarding this issue. As mentioned in the previous paragraphs, the expression of some genes depends on the stage of the disease or the age of the patients. It is clear from the data of the present study and the GSE5406 dataset that the patients participating in the present study were in the early stages of the disease, while the patients in the GSE5406 dataset were in the final stages. It can be the reason for the difference in *APP*, *MYH10*, and *MYH11* gene expressions in these two groups of patients. Moreover, the number of samples used in this study was limited; therefore, future studies can include larger groups of participants to investigate these genes’ status and identify other genes associated with the disease. Due to the limited available studies regarding the expression of these genes in IDCM patients, further studies are needed for a more detailed discussion. Finally, the lack of sample characteristics and clinical and survival information in the GSE5406 dataset were other limitations in this research.

## 5. Conclusions

In this study, seven genes that have increased expression in IDCM patients (obtained from bioinformatics analysis on the GSE5406 dataset) were selected for further investigation in patients referred to Jahrom Hospital. The RT-PCR results confirmed the overexpression of the *STAT1*, *IGF1*, *CCND1*, and *VEGFA* genes in blood samples of patients with IDCM, which was consistent with our GEO analysis. Among these four genes, *CCND1* was the only one not linked to IDCM or related heart disease in DisGeNET; therefore, according to our investigation, *CCND1* may be introduced as a potential biomarker for IDCM. Conversely, there were no significant differences in the expression of three DEGs (*APP*, *MYH10*, and *MYH11*) between patients and controls, which was inconsistent with GEO analysis. Considering the limited available studies on the expression of these genes in IDCM patients and the lack of detailed information on the GSE5406 dataset, more studies are needed for a more detailed discussion. Nevertheless, the present study may advance our understanding of IDCM pathogenesis and provide new targets for clinical diagnosis.

## Figures and Tables

**Figure 1 genes-14-01243-f001:**
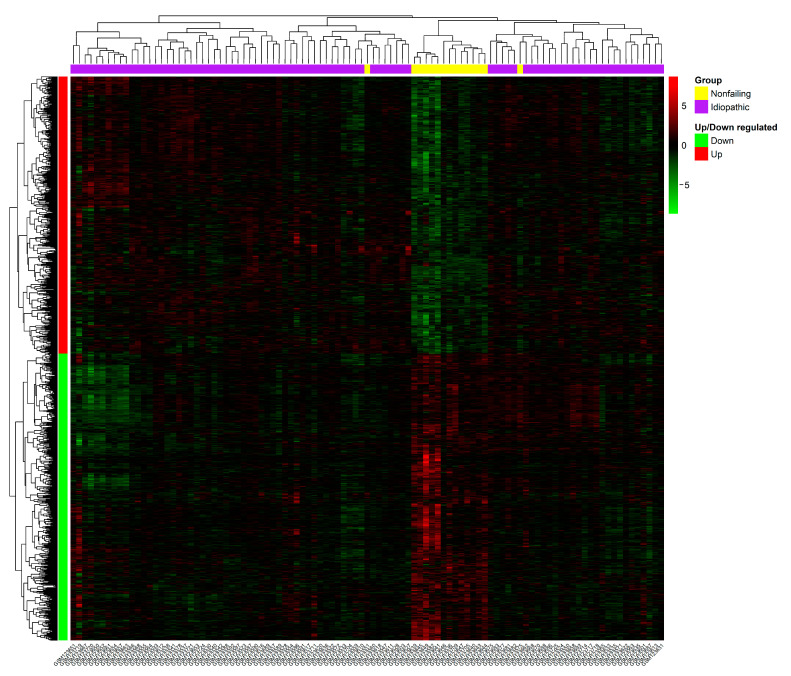
The hierarchical heatmap (based on Pearson correlation) of the DEGs. Each sample is represented by a column, and each gene is represented by a row. The change in color from green to red reflects the pattern of expression, from downregulation to upregulation.

**Figure 2 genes-14-01243-f002:**
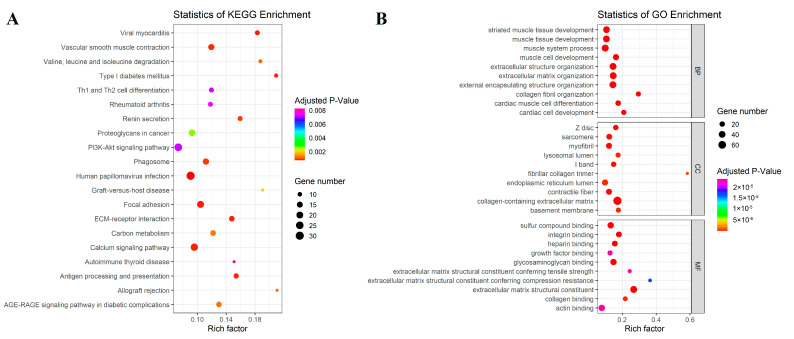
Scatter plot functional and pathway enrichment of upregulated DEGs. (**A**) Plot of the top 20 enriched KEGG pathways. (**B**) Plot of the top 10 enriched GO terms for molecular functions (MF), biological processes (BP), and cellular components (CC).

**Figure 3 genes-14-01243-f003:**
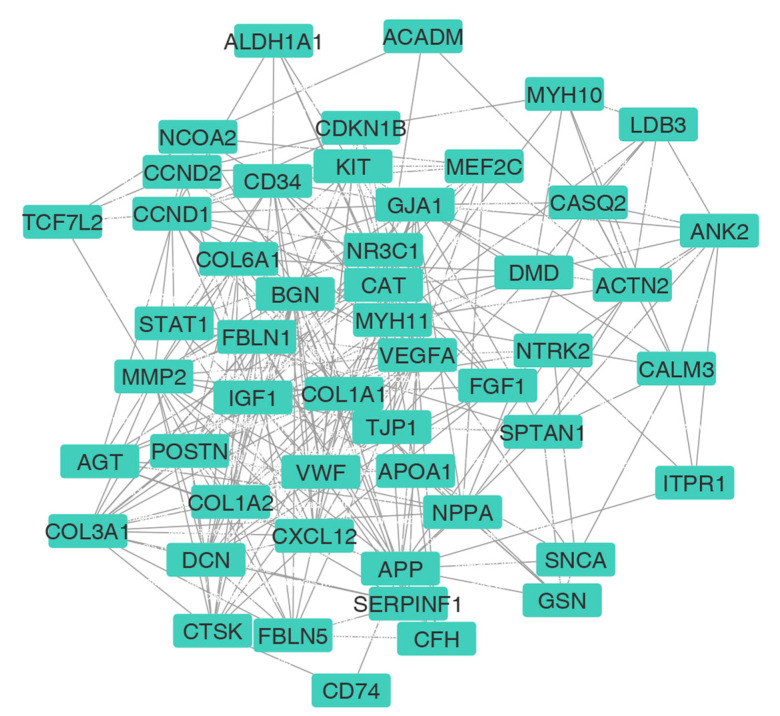
The protein–protein interaction (PPI) network of HUB genes.

**Figure 4 genes-14-01243-f004:**
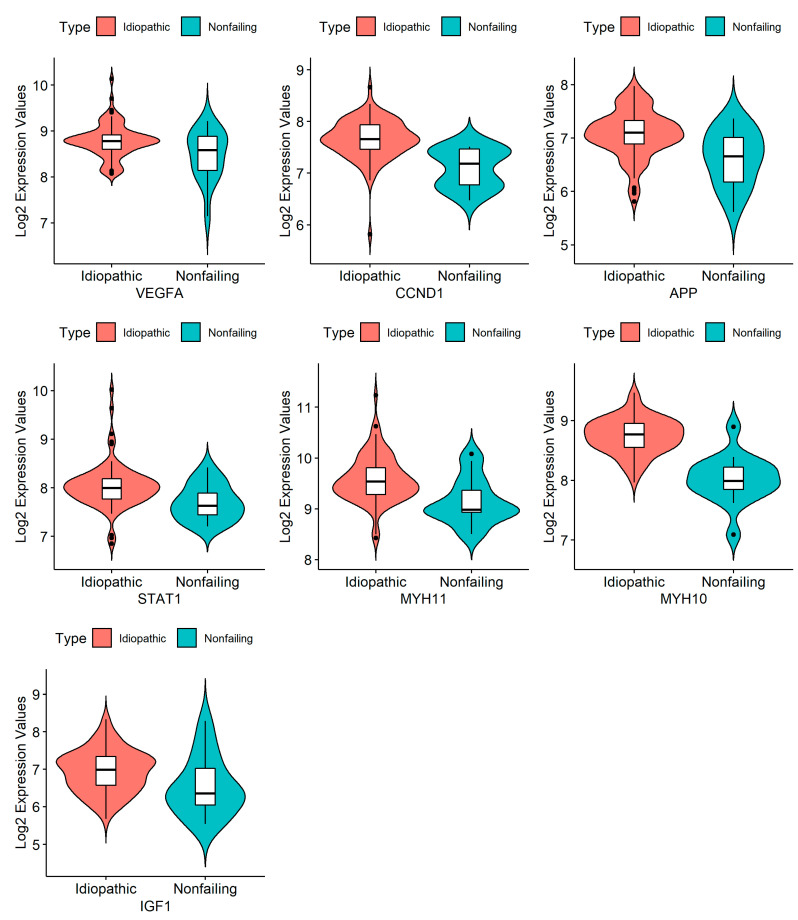
The violin plots of selected HUB of HUBs expression patterns. The vertical axes are not scaled.

**Figure 5 genes-14-01243-f005:**
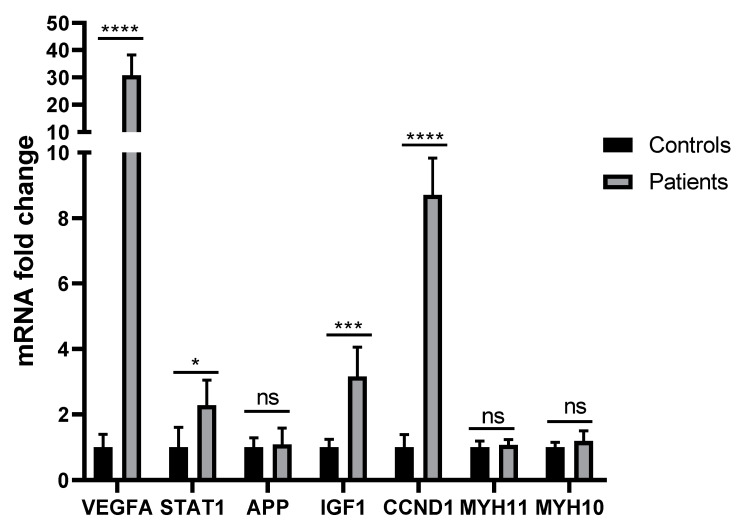
RT-PCR of the genes under study. Threshold cycle (Ct) values were obtained for each target and GAPDH as a control for normalization, and fold inductions were measured using the 2 ^−∆∆Ct^ method. * *p*-value < 0.05, *** *p*-value < 0.001, and **** *p*-value < 0.0001 vs. controls. “ns” stands for not-significant.

**Figure 6 genes-14-01243-f006:**
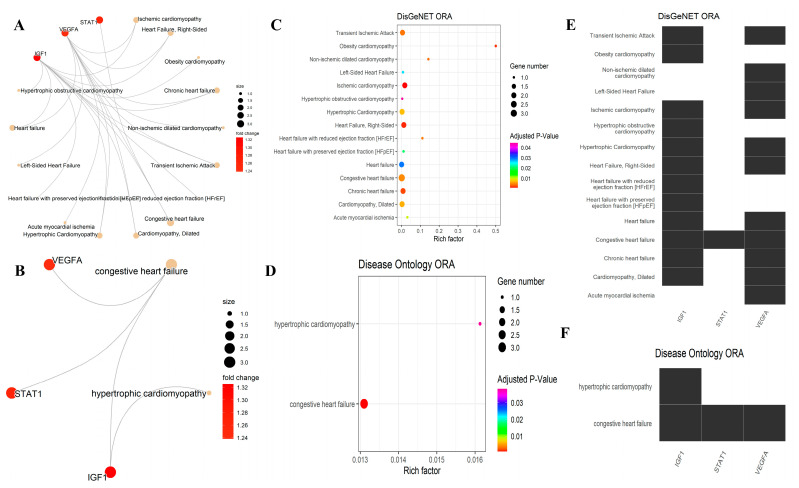
An overview of ORA for DisGeNET and DO generated by the “enrichplot” package. (**A**) and (**B**) gene-concept networks for DisGeNET and DO gene-disease associations, respectively. (**C**) and (**D**) dot plots illustrating these relationships with adj. *p*-value and Rich factor. (**E**,**F**) Heatmap-like functional classification representing the role and association of each gene among different diseases. Compared to all gene numbers involved in a disease term, the richest factor is the proportion of selected gene numbers (four HUBs). As the rich factor increases, the degree of disease increases.

**Table 1 genes-14-01243-t001:** The sequences of primers used in real-time PCR.

Primer		Sequence (5′→3′)
*VEGFA*	F	GGCAGAAGGAGGAGGGCAGAAT
	R	CATCGCATCAGGGGCACACA
*CCND1*	F	AGGCGGAGGAGAACAAACAGA
	R	TGAGGCGGTAGTAGGACAGGA
*MYH11*	F	GGACAAACTGCAAGCAAAGGTGA
	R	TTGTGTGAACCTCCCTGCTCT
*MYH10*	F	ATCTCGGCGTAAACTCCAGCG
	R	TCTGTGTCATCGTCGGAGAGC
*STAT1*	F	GTGGCAGGATGTCTCAGTGGT
	R	AACATCATTGGCAGCGTGCTC
*IGF1*	F	ACCATGTCCTCCTCGCATCTCT
	R	ACTGCTGGAGCCATACCCTGT
*APP*	F	TGGTGGGCGGTGTTGTCATAG
	R	GCCGTTCTGCTGCATCTTGGA
*GAPDH* (Housekeeping)	F	ATTATTCTCTGATTTGGTCGTAT
	R	CTCCTGGAAGATGGTGAT

**Table 2 genes-14-01243-t002:** Top 5 results of KEGG pathway enrichment analysis of DEGs.

ID	Description	Adjusted *p*-Value	Gene Symbol
hsa05165	Human papillomavirus infection	0.000771289	*ATP6V0E2/CCND1/CCND2/CDKN1B/COL1A1/COL1A2/COL6A1/COMP/EIF2AK2/FZD1/FZD7/HEY1/HLA-B/HLA-E/ITGB5/JAG1/LAMA4/LAMB1/LAMB2/MX1/NOTCH2/NOTCH3/PRKACB/STAT1/TCF7L2/THBS2/THBS4/TNXB/VEGFA/VWF*
hsa05416	Viral myocarditis	0.000833702	*CCND1/DMD/HLA-B/HLA-DMA/HLA-DMB/HLA-DPA1/HLA-DPB1/HLA-DRA/HLA-DRB1/HLA-E/SGCG*
hsa04510	Focal adhesion	0.000833702	*CCND1/CCND2/COL1A1/COL1A2/COL6A1/COMP/IGF1/ITGB5/LAMA4/LAMB1/LAMB2/MYLK3/PDGFC/PDGFD/PPP1R12B/THBS2/THBS4/TNXB/VEGFA/VEGFB/VWF*
hsa04512	ECM receptor interaction	0.000833702	*AGRN/COL1A1/COL1A2/COL6A1/COMP/ITGB5/LAMA4/LAMB1/LAMB2/THBS2/THBS4/TNXB/VWF*
hsa04020	Calcium signaling pathway	0.000833702	*ASPH/CALM3/CAMK2B/CASQ2/EDNRA/F2R/FGF1/GRM1/HRC/ITPR1/MYLK3/NTRK2/PDE1A/PDE1C/PDGFC/PDGFD/PLCB4/PLCE1/PRKACB/TPCN1/VDAC3/VEGFA/VEGFB*

**Table 3 genes-14-01243-t003:** Detailed information on ORA for DisGeNET and DO.

ID	Description	Gene Symbol	Adjusted *p*-Value	Database
C0349782	Ischemic cardiomyopathy	*IGF1/VEGFA*	0.000957133	DisGeNET
C0235527	Right-sided heart failure	*IGF1/VEGFA*	0.001407601	DisGeNET
C4552322	Obesity cardiomyopathy	*IGF1*	0.001552258	DisGeNET
C0264716	Chronic heart failure	*IGF1/VEGFA*	0.002243913	DisGeNET
C1168330	Non-ischemic dilated cardiomyopathy	*VEGFA*	0.003639825	DisGeNET
C0007787	Transient ischemic attack	*IGF1/VEGFA*	0.003972576	DisGeNET
C4509223	Heart failure with reduced ejection fraction (HFrEF)	*IGF1*	0.004298851	DisGeNET
C0018802	Congestive heart failure	*IGF1/STAT1/VEGFA*	0.004923199	DisGeNET
C0007193	Cardiomyopathy, dilated	*IGF1/VEGFA*	0.006671016	DisGeNET
C0007194	Hypertrophic cardiomyopathy	*IGF1/VEGFA*	0.007424915	DisGeNET
C0746731	Acute myocardial ischemia	*VEGFA*	0.009968188	DisGeNET
C4509226	Heart failure with preserved ejection fraction (HFpEF)	*IGF1*	0.020442736	DisGeNET
C0023212	Left-sided heart failure	*VEGFA*	0.026246253	DisGeNET
C0018801	Heart failure	*IGF1/VEGFA*	0.029611829	DisGeNET
C4551472	Hypertrophic obstructive cardiomyopathy	*IGF1*	0.044684056	DisGeNET
DOID:6000	Congestive heart failure	*IGF1/STAT1/VEGFA*	0.000875442	DO
DOID:11984	Hypertrophic cardiomyopathy	*IGF1*	0.038275847	DO

## Data Availability

No new data were created or analyzed in this study. Data sharing does not apply to this article.

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
