# Peer review of "CCND1 Overexpression in Idiopathic Dilated Cardiomyopathy: A Promising Biomarker?"

_genes, 2023, doi:10.3390/genes14061243_

Round 1

Reviewer 1 Report

This is a difficult paper to follow.  The title itself is entirely misleading:  at best CCND1 could be a marker for IDCM,  never a therapy.

The study consists as two parts, although in the text they are not clearly distinguished.

Firstly a bioinformatic study of an existing gene expression dataset using cardiac tissue is performed.

Secondly an RNA expression study of plasma from the authors own patient cohort.  It is far from clear whether the conclusions from one can be applied to the other.

study of DEGs from GSE5406:  the authors should acknowledge and describe the nature and origin of the dataset:  Hannenhalli et al. Circulation. 114 (2006) 1269–1276. DOI10.1161/circulationaha.106.632430. They should comment on the criteria for the IDCM diagnosis used.  The cohort used likely includes both acquired IDCM and facial IDCM cases-is there any data on this?  There are likely to be further heterogeneities in the court e.g in rersponse to adrenergic stimulation.   The authors should look at their results with this in mind:  there may be two or more clusters of samples with different characteristics- this would be very interesting since treating all samples as equal may give a false result.

The analysis is largely conventional and the results should be compared with previous studies using this dataset which came up with different results.  I am not expert enough but I do feel that the justification given lines 326-343 is rather one-sided.

For the second study using patient serum rather than heart muscle, five genes only are studied.  The reasons for selecting them are rather opaque and yet are crucial to the rest of the paper.  The gene set seems a bit arbitrary and a lot more justification is needed here.

The RNA quantification of these genes in patient plasma seems straightforward, however it is not at a all clear if the selection criteria for the IDCM diagnosis matches the criteria for the GSE5406 dataset previously studied.  Very substantial increase in VEGFA and CCND1 are found in plasma according to figure 5 but the figure lacks a clear descriptive legend.  Table 3 is unclear:  is it the same data as in Figure 5?  Is it just the IDCM data?  If so where is the non-failing data?  The column "Expression"  is not defined here.  "standard error seems to show a range rather than a standard error., likewise for 95% CI.  "CI" is not defined so this table is very hard to read.   If it just restates figure 5 it is redundant anyway.

A third calculation uses dataset DisGeNET  and DO,  which are largely unexplained.  Do they contain data relevant to heart?  The data mining failed to give any "cardio-related experimental evidence for CCND1" (presumably you mean no correlation of IDCM and increased CCND1.)

So overall, the evidence for increased CCND1 rest on the patient cohort study using plasma.  Is there any reason at all to assert that the CCND1 is cardiac-derived, like troponin?  This would be interesting perhaps if, as the authors assert it is IDCM-specific, but I do not see any evidence that test this hypothesis.

The paper is thus confusing and the conclusions are unclear and not backed up with credible evidence.

Moderate editing of English language

Reviewer 2 Report

The authors studied the gene expression profiles from IDCM patients and found the VEGFA and CCND1 are the most significant upregulated among multiple hub genes.

This study has clear methodology description and showed interesting results about different genes and association with IDCM dieases. Here are some minor suggestions:

1. The labeling in Figure 1 was barely to see, as it is a huge data set, I wonder if there is any better way to improve the resolution of the heatmap.

2. The finding of significant increase of CCND1 seems out of the expectation, which is interesting, but in the current study, the authors only found the RNA expression change in the patients blood samples, is it too early to say in the title "a promising target for future therapies"?

Reviewer 3 Report

The study highlights the importance of further investigations involving a larger cohort of patients and additional genes to strengthen the results. Nonetheless, the findings contribute to our understanding of the genetic mechanisms underlying IDCM and offer potential targets for future therapeutic interventions.

Overall, this study provides valuable insights into the pathogenesis of IDCM and identifies potential biomarkers. The researchers employed robust methods, including bioinformatics analysis and RT-PCR validation, to investigate gene expression in IDCM patients. The confirmation of overexpression for STAT1, IGF1, CCND1, and VEGFA genes in IDCM patients is consistent with the analysis of the GEO dataset. Notably, the finding of CCND1 as a potential biomarker for IDCM, despite its lack of association with the disease in DisGeNET, is an interesting observation.

However, there are a few limitations that need to be addressed. The sample size used in this study is limited, warranting caution in generalizing the results. The discrepancy between the RT-PCR results and the GEO analysis regarding the expression of APP, MYH10, and MYH11 genes raises concerns. Additionally, the lack of sample characteristics, clinical information, and survival data in the NCBI dataset is a drawback. Consideration should also be given to the potential influence of disease stage and patient age on gene expression.

Taking these points into account, this study presents promising findings but requires further validation with larger cohorts and more comprehensive clinical data.

normal 

Round 2

Reviewer 1 Report

This manuscript has been improved with proper explanations of previously obscure or ambiguous points.

The conclusions have been suitably toned-down so that they now correspond to the results.

OK